# Clinical Trial on the Safety and Tolerability of Personalized Cancer Vaccines Using Human Platelet Lysate-Induced Antigen-Presenting Cells

**DOI:** 10.3390/cancers15143627

**Published:** 2023-07-14

**Authors:** Terutsugu Koya, Kenichi Yoshida, Misa Togi, Yo Niida, Sumihito Togi, Hiroki Ura, Shuichi Mizuta, Tomohisa Kato, Sohsuke Yamada, Takeo Shibata, Yi-Chang Liu, Shyng-Shiou Yuan, Deng-Chyang Wu, Hirohito Kobayashi, Taiju Utsugisawa, Hitoshi Kanno, Shigetaka Shimodaira

**Affiliations:** 1Department of Regenerative Medicine, Kanazawa Medical University, Kahoku 920-0293, Ishikawa, Japan; koya@kanazawa-med.ac.jp (T.K.); m-togi@kanazawa-med.ac.jp (M.T.); 2Center for Regenerative Medicine, Kanazawa Medical University Hospital, Kahoku 920-0293, Ishikawa, Japan; ken1-y@kanazawa-med.ac.jp; 3Division of Genomic Medicine, Department of Advanced Medicine, Medical Research Institute, Kanazawa Medical University, Kahoku 920-0293, Ishikawa, Japan; niida@kanazawa-med.ac.jp (Y.N.); togi@kanazawa-med.ac.jp (S.T.); h-ura@kanazawa-med.ac.jp (H.U.); 4Department of Hematology and Immunology, Kanazawa Medical University, Kahoku 920-0293, Ishikawa, Japan; mizuta@spacelan.ne.jp; 5Division of Stem Cell Medicine, Department of Advanced Medicine, Medical Research Institute, Kanazawa Medical University, Kahoku 920-0293, Ishikawa, Japan; tkato@kanazawa-med.ac.jp; 6Department of Pathology and Laboratory Medicine, Kanazawa Medical University, Kahoku 920-0293, Ishikawa, Japan; sohsuke@kanazawa-med.ac.jp; 7Department of Obstetrics and Gynecology, Kanazawa Medical University, Kahoku 920-0293, Ishikawa, Japan; ma30408@kanazawa-med.ac.jp; 8Division of Hematology and Oncology, Department of Internal Medicine, Kaohsiung Medical University Hospital, Kaohsiung 807, Taiwan; ycliu@kmu.edu.tw; 9Faculty of Medicine, College of Medicine, Kaohsiung Medical University, Kaohsiung 807, Taiwan; 10Office of Research & Development, Kaohsiung Medical University, Kaohsiung 807, Taiwan; yuanssf@kmu.edu.tw; 11Internal Medicine, School of Medicine, Kaohsiung Medical University, Kaohsiung 807, Taiwan; dechwu555@gmail.com; 12Division of Transfusion and Cell Therapy, Tokyo Women’s Medical University, Adachi Medical Center, Adachi 123-8558, Tokyo, Japan; kobayashi.hirohito@twmu.ac.jp; 13Department of Transfusion Medicine and Cell Processing, Tokyo Women’s Medical University, Shinjuku 162-8666, Tokyo, Japan; utsugisawa.taiju@twmu.ac.jp (T.U.); kanno.hitoshi@twmu.ac.jp (H.K.)

**Keywords:** personalized cancer vaccine, in silico prediction, antigen-presenting cells, enzyme-linked immunosorbent assay, memory T cells

## Abstract

**Simple Summary:**

In this study, we developed human platelet lysate-induced antigen-presenting cells (HPL-APCs) from peripheral monocytes with a high potency of presentation ability. This study aimed to verify the safety, tolerability, and immunoinducibility of HPL-APCs loaded with cancer candidate antigens as a Phase I study. As a result of the interim analysis, safety and tolerability were confirmed in three enrolled patients, and the immune response to cancer antigen candidate peptides predicted in silico was confirmed in two completed cases. This clinical study is the first to verify the feasibility and immunoinducibility of a personalized cancer vaccine using HPL-APCs that would be expected to demonstrate further antitumor activity through optimized combination therapies.

**Abstract:**

Research and development of personalized cancer vaccines as precision medicine are ongoing. We predicted human leukocyte antigen (HLA)-compatible cancer antigen candidate peptides based on patient-specific cancer genomic profiles and performed a Phase I clinical trial for the safety and tolerability of cancer vaccines with human platelet lysate-induced antigen-presenting cells (HPL-APCs) from peripheral monocytes. Among the five enrolled patients, two patients completed six doses per course (2–3 × 10^7^ cells per dose), and an interim analysis was performed based on the immune response. An immune response was detected by enzyme-linked immunosorbent spot (ELISpot) assays to HLA-A*33:03-matched KRAS^WT^, HLA-DRB1*09:01-compliant KRAS^WT or G12D^, or HLA-A*31:01-matched SMAD4^WT^, and HLA-DRB1*04:01-matched SMAD4^G365D^ peptides in two completed cases, respectively. Moreover, SMAD4^WT^-specific CD8^+^ effector memory T cells were amplified. However, an attenuation of the acquired immune response was observed 6 months after one course of cancer vaccination as the disease progressed. This study confirmed the safety and tolerability of HPL-APCs in advanced and recurrent cancers refractory to standard therapy and is the first clinical report to demonstrate the immunoinducibility of personalized cancer vaccines using HPL-APCs. Phase II clinical trials to determine immune responses with optimized adjuvant drugs and continued administration are expected to demonstrate efficacy.

## 1. Introduction

Advances in next-generation sequencing technology make it possible to treat cancer as precision medicine based on individual cancer genome information. However, only a limited number of patients can benefit from individualized therapy [1], and cancer vaccines based on patient-specific cancer genome information are expected as novel cancer treatments [2].

Cancer vaccines targeting either tumor-associated antigens or tumor-specific antigens, including neoantigens generated by genetic mutations of cancer cells, are being developed [3]. Among cancer vaccine platforms including RNA, DNA, peptides, and dendritic cells (DCs) [4], DC vaccines have been reported to have strong immunoinductibility [5,6]. Immune memory is important for the sustained efficacy of cancer vaccines [7], and the function of DCs that play a central role in immune acquisition is attracting attention [8].

Clinical trials of neoantigen-pulsed DC vaccines using in silico analysis based on cancer genomic information in patients with advanced lung cancer have been conducted [9], and the immune response to neoantigens targeting compatible peptides on HLA class I or II has been confirmed. On the contrary, the immune memory induced by DC vaccination must be comprehensively investigated.

Clinical trials have been conducted on classical DC vaccines derived from monocytes using granulocyte-macrophage colony-stimulating factor (GM-CSF) and interleukin-4 (IL-4) [10,11,12]. GM-CSF and interferon-α-generated DCs (IFN-DCs) are reported to exhibit antigen presentation and termination activity against tumor cells via TRAIL, FAS ligand and granzyme B/ perforin pathway [13,14]. We further developed IFN-DCs using human platelet lysates (HPL), named human platelet lysate-inducible antigen-presenting cells (HPL-APCs) with high endocytic and proteolytic activities [15], which exhibit strong antigen presentation ability compared to IFN-DCs [15]. We detected SMAD4^P130L^ in pancreatic cancer cells in malignant pleural effusion and reported the inducibility of HLA-A*11:01-restricted SMAD4^P130L^ neoantigen-specific CD8^+^ T cells in vitro using HPL-APCs [16]. Therefore, HPL-APCs are expected to be used as a platform for personalized cancer vaccines.

In this study, we verified the safety and tolerability of HPL-APCs loaded with personalized cancer candidate antigens based on cancer genomic profiles and immune memory in a Phase I trial in advanced and recurrent cancers.

## 2. Materials and Methods

### 2.1. Manufacture of HPL-APCs

Peripheral blood mononuclear cell (PBMC)-rich fractions through leukapheresis were harvested using a Spectra Optia^®^ cell separator (Terumo BCT, Inc., Tokyo, Japan). PBMCs were separated with density gradient centrifugation using Ficoll-Plaque Premium (Global Life Sciences Solutions USA LLC, Marlborough, MA, USA) and were washed out with phosphate-buffered saline (Wako Pure Chemicals Ltd., Osaka, Japan). Then, the PBMCs were suspended in serum-free medium DCO-K (Nissui Pharmaceutical Co., Ltd., Tokyo, Japan) supplemented with 5% of GMP graded UltraGRO^TM^-PURE GI (hereinafter called “HPL”; AventaCell BioMedical Corp., Atlanta, GA, USA), which is an HPL reagent approved as a material for regenerative medicine products by the Pharmaceuticals and Medical Devices Agency in Japan. The PBMCs’ suspension was seeded into adherent culture dishes. After 30 min incubation, the nonadherent cells were removed, and the adherent monocytes were cultured in DCO-K medium containing 100 ng/mL recombinant human GM-CSF (Miltenyi Biotec B.V. & Co. KG, Bergisch Gladbach, Germany) and 1 µg/mL PEGylated-IFN-α-2a (PEGASYS^®^; Chugai Pharmaceutical Co., Ltd., Tokyo, Japan) with 5% HPL for 3-day cultivation. For maturation, immature APCs were collected and resuspended in DCO-K medium with 5% HPL with 100 ng/mL GM-CSF, 1 µg/mL PEGASYS^®^, 10 µg/mL OK-432 (Streptococcal Preparation, Chugai), 10 ng/mL prostaglandin E2 (PGE2; Kyowa Pharma Chemical Co., Ltd., Toyama, Japan), and 10 µg/mL cancer antigen candidate peptides (synthesized by GenScript USA Inc., Piscataway, NJ, USA). After cultivation for 24 h in low adherent cell culture dishes, the cells were harvested and cryopreserved at −152 °C. The phenotype of HPL-APCs was determined using BD FACSLyric^TM^ (BD Biosciences, Franklin Lakes, NJ, USA). HPL-APCs were treated with FcR Blocking Reagent (Miltenyi Biotec) at room temperature for 10 min and stained with antibodies (Appendix A) at 4 °C for 60 min. Their respective isotype antibodies were used as the negative control. Live cells defined as negative for 7-aminoactinomycin D (7-AAD, Miltenyi Biotec) were analyzed using FACSLyric^TM^ (BD Biosciences). Data were analyzed with FlowJo^TM^ (BD Biosciences).

### 2.2. Prediction of Cancer Antigen Candidate Peptides

Cancer antigen candidate peptides based on the binding affinities of 8- to 14-mer to HLA-A or 12- to 18-mer to HLA-DPB1/DRB1 were examined using the IEDB analysis resource NetMHCpan (ver. 4.1) [17]. The peptides that predicted the binding affinity the most strongly were selected for the HPL-APC vaccine according to the following priority order: (1) mutated peptides with IC_50_ < 500 nM and (2) wild-type peptides with IC_50_ < 500 nM.

### 2.3. HPL-APC Vaccination 

The release criteria of the HPL-APC vaccine were set at CD86^+^HLA-DR^+^, >70% viability, negative for bacterial and fungal infections, < 1 EU/mL of endotoxin, and negative for mycoplasma. Cryopreserved HPL-APCs were thawed during each session and suspended in 1 mL of saline. The HPL-APC vaccine was intradermally and bilaterally administered near the axillary region and groin. 

### 2.4. Evaluation of Safety and Tolerability 

The evaluation of safety included the following: (1) any systemic reactions at an early phase within 48 h of the intradermal injection of the HLA-APC vaccine (presence of high fever, hypotension, shock tachycardia, bradycardia, breathing difficulties, or skin rash) and (2) local skin reactions at the injection sites; other symptoms, including nausea, vomiting, diarrhea, appetite loss, mucosal ulcer, or central nervous system disorders; and laboratory test dysfunctions: anemia, reduced white blood cells, reduced platelet count, or elevated kidney and liver serum levels during or after the course of vaccination as described previously [10]. 

### 2.5. Enzyme-Linked Immunosorbent Spot (ELISpot) Assays 

To investigate the antigen-specific IFN-γ production, ELISpot assays were performed using the human IFN-γ ELISpot PLUS kit (Mabtech AB, Nacka Strand, Sweden). Moreover, 1 × 10^6^ PBMCs were incubated in 96 wells with AIM-V medium with 10% fetal bovine serum (FBS) and 10 μM peptides. After 16–20 h of incubation, IFN-γ producing cells were detected according to the manufacturer’s protocol. The spots were calculated using an automatic ELISpot reader (AID ELISpot Reader Classic ELR 07; Autoimmun Diagnostika GmbH, Strassberg, Germany). The counted numbers with cancer candidate peptides were compared with the HLA-matched HIV peptide (Appendix A) as a negative control. The immune response to cancer antigen candidate peptides was defined according to the following criteria: (1) at least 15 spots per 1 × 10^6^ PBMCs and (2) at least 1.5-fold more presence of spots than negative control peptide spots [18]. 

### 2.6. Detection of Monocytic MDSCs (M-MDSCs) and Regulatory T Cells (Tregs)

PBMCs were treated with FcR Blocking Reagent (Miltenyi Biotec) and stained with antibodies in Appendix A or Appendix A. Live cells defined as negative for 7-AAD (Miltenyi Biotec) were analyzed using FACSCanto II (BD Biosciences) or FACSLyric (BD Biosciences). Tregs were defined as CD45^+^CD3^+^CD4^+^CD25^+^CD127^−/low^ cells [19]. M-MDSCs were defined as CD45^+^CD11b^+^CD33^+^CD14^+^HLA-DR^−^ cells [20]. 

### 2.7. Detection of Antigen-Specific Memory T Cell Subsets

PBMCs were incubated with an AIM-V medium with 10% FBS and 10 μM peptides. HLA-matched HIV peptide was used as the negative control. After 16–20 h incubation, cells were washed and treated with FcR Blocking Reagent (Miltenyi Biotec) and stained with antibodies as shown in Appendix A at 4 °C for 60 min. To detect antigen-specific memory T cells, the activation of CD137 on CD8^+^ T cells was analyzed [21]. Further, naïve cells as CD45RO^−^CCR7^+^, central memory cells as CD45RO^+^CCR7^+^, effector memory cells as CD45RO^+^CCR7^−^, and effector cells as CD45RO^−^CCR7^−^ were identified for memory T cell subsets [22]. 

### 2.8. Case Report 

#### 2.8.1. Case 1 (Patient 2)

A male patient in his 50s was diagnosed with pancreatic cancer (stage IV, KRAS^G12D^ gene mutation). Apheresis was performed without adverse events such as vasovagal reflexes. A course of HPL-APC vaccination for KRAS combined with gemcitabine + protein-bound paclitaxel chemotherapy was completed in six doses, with a total cell count of 14.0 × 10^7^ cells (mean viable cell rate, 91.9%). No allergic reaction with delayed-type hypersensitivity (DTH) lower than 5 mm to peptides was noted, and a grade 2 fever reaction and redness of grade 2 at the vaccine-injected sites were observed; however, no other adverse reactions or adverse events were reported. No adverse events were found during the observation period, and follow-up continued. Diagnostic imaging after one course revealed a stable disease (SD); however, 4 months after one course, imaging of the metastasized lung showed a progressive disease (PD), and chemotherapy with 5FU + irinotecan hydrochloride hydrate was performed.

#### 2.8.2. Case 2 (Patient 3)

A female patient in her 40s was diagnosed with cervical cancer (adenocarcinoma, stage IV, KRAS^G12D^ and SMAD4^G365D^ gene mutations). Apheresis was performed with no adverse events. One course of the HPL-APC vaccine against KRAS and SMAD4 in combination with oral cyclophosphamide was completed, with a total cell count of 17.7 × 10^7^ cells (mean viable cell rate, 93.9%) without any quality problems. No allergic reactions with DTH reaction of peptides lower than 5 mm were noted. Grade 2 fever reactions, chills, arthralgia, and redness of grade 2 in the vaccination area were observed but were well tolerated. At the second dose, vaginal bleeding due to primary disease infiltration was observed; however, no other vaccine-related adverse reactions or adverse events occurred. Diagnostic imaging after one course was SD; however, at 4 months after one course, a colostomy was performed because of cancer progression.

#### 2.8.3. Case 3 (Patient 5)

A female patient in her 70s was diagnosed with pancreatic cancer (postoperative recurrence stage IV, KRAS^G12D^ and TP53^E258K^ gene mutations). Apheresis had already been performed in the WT1 clinical study to obtain materials. During 5FU + irinotecan hydrochloride hydrate salvage therapy, she was administered the HPL-APC vaccine against KRAS and TP53. Neither allergic reaction to DTH nor febrile reactions occurred; however, no adverse reactions or adverse events other than redness of grade 1 were found at the vaccination site. Because the cancer progressed to aortic lymph node metastasis, multiple liver metastases, bile duct obstruction, and portal vein invasion, only bile duct stents and conservative treatment including celiac nerve block were administered without anticancer drugs. After four vaccine lots were administered, she died from the cancer’s further progression.

## 3. Results

### 3.1. Preparation of Personalized HPL-APC Vaccine with Cancer Antigen Candidate Peptides

By targeting gene products with pathogenic mutations from cancer genomic information, we predicted the affinity of cancer antigen candidate peptides for HLA typing in each patient. Peptides exhibiting an affinity of IC50 < 500 nM were selected for HLA class I and HLA class II (Table 1 and Table 2). In patients 2 and 3, killer peptides for the activation of CD8^+^ T cells with a high affinity for HLA class I of IC_50_ < 50 nM were identified (KRAS^WT^-HLA-A*33:03, IC_50_ = 11 nM; SMAD4^WT^-HLA-A*31:01, IC_50_ = 6 nM). Three enrolled patients had HLA-DRB1*09:01 and were identified to have helper peptides for the activation of CD4^+^ T cells with a moderate affinity for KRAS^G12D^ of IC_50_ < 500 nM (IC_50_ = 181 nM, KRAS^G12D^-HLA-DRB1*09:01). The quality of the HPL-APC vaccine harboring candidate peptides targeting personalized cancer antigens is verified in Table 3. HPL-APCs showed high yields (>23%), viability (>94%), and purity (>91%) in three patients. Phenotypic analysis using a flow cytometer revealed HLA-ABC and HLA-DR expression with costimulatory molecules CD40, CD80, and CD86. These results confirmed the validated quality of HPL-APCs containing cancer antigen-specific peptides for use in clinical research.

### 3.2. Detection of Immune Responses to Personalized Cancer Antigen Candidate Peptides

The interim analysis of immune responses was performed on three enrolled participants, including two who completed six doses per course. Immune responses were monitored by ELISpot assays using personalized cancer antigen candidate peptides (Figure 1). The immune response was evaluated according to the criteria for ELISpot assays [18]. In patient 2, an immune response to HLA-A*33:03-matched KRAS^WT^ was detected after six doses of the HPL-APC vaccine (Figure 1b). Immune responses to KRAS^WT^ peptide compatible with HLA-DRB1*07:01 and KRAS^G12D^ to HLA-DRB1*09:01 were detected (Figure 1c). Furthermore, an immune response was detected using a KRAS^WT^-HLA-DRB1*09:01 that had a sequence corresponding to KRAS^G12D^-HLA-DRB1*09:01. These immune responses attenuated 6 months after completion of HPL-APC vaccination. In patient 3 (Figure 1d–f), an immune response to HLA-A*31:01-matched SMAD4^WT^ peptide was strongly detected after three and six doses of the HPL-APC vaccine (Figure 1e). Positive reactions to SMAD4^G365D^ peptide compatible with HLA-DRB1*04:01 and those of KRAS^G12D or WT^ to HLA-DRB1*09:01 were detected (Figure 1f). Six months after one course of the HPL-APC vaccine, the immune response was also diminished. An interim analysis of patient 5 after three doses of the HPL-APC vaccine (Figure 1g,h) showed no immune response. When the immunosuppressive factors of Tregs and M-MDSCs were examined (Appendix A), a marked increase in M-MDSCs was observed in patient 5. On the contrary, patients 2 and 3 did not show quantitative fluctuations in Tregs and M-MDSCs.

### 3.3. Detection of SMAD4^WT^ HLA-A*31:01-Specific CD8^+^ Effector Memory T Cells

Immunoinductibility of the HPL-APC vaccine was verified from the viewpoint of immune memory. Memory subsets of CD8^+^ T and CD4^+^ T cells in PBMCs during HPL-APC vaccination were analyzed (Appendix A). Patient 2, who was positive in the ELISpot assays, showed an increase in CD8^+^ T effector memory T cells (T_EM_) after three and six doses of the HPL-APC vaccine (before vaccination, 19.6%; after vaccination #3, 26.6%; and after vaccination #6, 27.3% in CD8^+^ T cells). Patient 3 also showed an increase in CD8^+^ T_EM_ after HPL-APC vaccination (before vaccination, 29.3%; after vaccination #3, 43%; and after vaccination #6, 46.5% in CD8^+^ T cells). However, a significant decrease in T_EM_ was observed 6 months after six doses (6 months from vaccination #6, 25.2% in patient 3). In addition, changes in cancer antigen candidate peptide-specific CD8^+^ memory T cells were verified. With antigen candidate peptides of KRAS^WT^-HLA-A*33:03 or SMAD4^WT^-HLA-A*31:01, CD137 expression in CD8^+^ T cells was analyzed (Figure 2). Patient 2, after six doses of the HPL-APC vaccine, showed few differences in CD137 expression due to KRAS^WT^-HLA-A*33:03 compared with an HIV-negative peptide (CD137^+^CD8^+^ T cells: HIV-HLA-A*33:03, 2.09% vs. KRAS^WT^-HLA-A*33:03, 2.24% in Figure 2a). In patient 3, SMAD4^WT^-HLA-A*31:01 increased CD137 expression (CD137^+^CD8^+^ T cells: HIV-HLA-A*31:01, 0.46%; SMAD4^WT^-HLA-A*31:01, 1.14% in Figure 2b). T_CM_ in CD137^+^CD8^+^ T cells decreased, and an increase in T_EM_ was observed (T_CM_: before vaccination vs. after vaccination #6, 61.5% vs. 43.8%; T_EM_: before vaccination vs. after vaccination #6, 38.5% vs. 56.2%).

## 4. Discussion

This study confirmed the safety and tolerability of the HPL-APC vaccine in two patients with pancreatic cancer (patients 2 and 5) and one with cervical cancer (patient 3) and clarified the immunoinductibility to personalized cancer antigen candidate peptides using ELISpot assays. HLA class I-compatible killer peptides or HLA class II-compliant helper peptides targeting KRAS, SMAD4, or TP53 were predicted from cancer genomic profiles (Table 2), which are pancreatic cancer driver genes frequently identified as pathogenic mutations [23]. Immunoinductibility to KRAS^WT or G12D^ peptides for HLA class II was confirmed in patient 2 after six sessions, and that to SMAD4^WT^ peptide for HLA class I and SMAD4^G365D^ peptide for HLA class II in patient 3 were detected after three doses (Figure 1). Although these peptides for the induction of antigen-specific immune response to CD8^+^ and CD4^+^ T cells in PBMCs after the HPL-APC vaccine were suggested, future analysis is needed to clarify the responding cells from IFN-γ production using flow cytometry. The activation of cancer antigen-specific CD8^+^ T and CD4^+^ T cells is a critical issue for the efficacy of cancer vaccines [7]. KRAS is the most frequently mutated isoform present in 22% of all tumors [24], and KARS^G12D^ is the highest frequency of approximately 35% among them [25]. Immunoinductibility could not be observed for KRAS^G12D^-HLA-A*11:01 (VVGADGVGK, 9-mer peptide) in patient 3 (Figure 1d). However, as the TCR repertoire responding to VVVGADGGGK (10-mer peptide) to HLA-A*11:01 binding KRAS^G12D^ peptide was reported [26], the immunoinductibility of HPL-APCs to these peptides must be verified in the future with an additional number of cases. On the contrary, KRAS^G12D^-HLA-DRB1*09:01 helper peptide common to all three cases was chosen (TEYKLVVVGADGVGK, 15-mer peptide in Table 2). Although a TCR repertoire for HLA-DRB1*08:01 binding KRAS^G12D^ was reported [27], immunoinduction against KRAS^G12D^-HLA-DRB1*09:01 helper peptide was first identified; however, the TCR repertoire has not yet been analyzed. Interestingly, an immune response was also detected when using a KRAS^WT^-HLA-DRB1*09:01 that had a sequence corresponding to KRAS^G12D^-HLA-DRB1*09:01 (Figure 1c,f). The TCR repertoire analysis induced by the KRAS^G12D^-HLA-DRB1*09:01 helper peptide is expected to reveal specificity and further verify the differences in the antitumor response.

DC vaccines can enhance effector and memory T cell responses, which are important for antitumor immunity and have been used in clinical trials of cancer immunotherapy [28,29]. A Phase I clinical trial of a DC vaccine pulsed with Wilms’ tumor 1 (WT1)-specific MHC class I/II-restricted epitope for pancreatic cancer in combination with chemotherapy was reported. An association was found between WT1/HLA-A*24:02-specific memory cells in CD8^+^ T cells and overall survival (OS) [12]. WT1-specific central and effector memory CD8^+^ T cells were observed in cancerous pleural effusion in a patient with long-survival pancreatic cancer after WT1-pulsed DC vaccination [16]. In Phase II trials to evaluate the efficacy of personalized cancer vaccines using HPL-APCs, the detection of antigen-specific memory T cells could be a biomarker of therapeutic prognosis. In patients 2 and 3, CD8^+^ T_EM_ increased after three doses of the HPL-APC vaccine (Appendix A), and in patient 3, an increase in antigen-specific CD8^+^ T_EM_ was observed with SMAD4^WT^-HLA-A*31:01 peptide in a small number of cells due to limited clinical specimens (Figure 2b). Because T_EM_ has strong cytotoxicity with high cytokine production capacity [30], it was consistent with the increase in IFN-γ-producing cells observed with SMAD4^WT^ HLA-A*31:01 peptide (Figure 1e). These results confirmed the ability of HPL-APCs to enhance antigen-specific CD8^+^ memory T cells. On the contrary, the detection of CD8^+^ memory T cells to KRAS^WT^-HLA-A*33:03 and that of CD4^+^ memory T cells responding to KRAS^WT or G12D^ or SMAD4^G365D^ helper peptides remain to be elucidated. A method for highly efficient detection of neoantigen-responsive T cells and identification of these TCRs has been reported for personalized cancer immunotherapy [31]. Cancer antigen-specific memory T cells induced by the HPL-APC vaccine using a method that can be easily detected from the peripheral blood via TCRs as an indicator must be identified.

In patients 2 and 3, who completed the HPL-APC vaccination, the acquired immune response was attenuated 6 months after HPL-APC vaccination (Figure 1b,c,e,f), and in patient 2, a significant decrease in T_EM_ was observed (Appendix A). In patients with long-term survival receiving continuous WT1-DC vaccine, acquired WT1-CTLs were maintained [32], and booster effects of additional vaccines were observed [33]. As DC vaccines have been expected to have antitumor effects by priming and boosting [34], the immune memory acquired by the HPL-APC vaccine must be also preserved. Therefore, activating the immune response with additional administrations of the HPL-APC vaccine is expected. As the life span of DC in the lymph nodes is limited to a few days [35,36], the persistence of the HPL-APC vaccine also needs to be monitored in patients to ensure long-lasting efficacy. On the contrary, the immunosuppressive factor M-MDSCs increased, and no immune response was observed in patient 5 (Figure 1g,h). The combination of DC vaccines with chemotherapy is important to create a state favoring expansion of antitumor effector cells, and inhibition, depletion, or both, of Tregs and MDSCs [37]. In a future Phase II study to verify the efficacy of the HPL-APC vaccine, the antitumor immune reactions in combination with anticancer drugs that eliminate or abolish the function of M-MDSCs and immune checkpoint inhibitors that restore exhausted T cells must be optimized [38].

## 5. Conclusions

This is the first Phase I clinical trial that demonstrates the safety, tolerability, and immunoinductibility of the HPL-APC vaccine with personalized cancer antigen candidate peptides. The HPL-APC vaccine can enhance antigen-specific CD8^+^ memory T cells. The maintenance of immune memory might require continued HPL-APC vaccination and improvement of the immunosuppressive environment. The feasibility of personalized cancer vaccines using HPL-APCs has been clarified, and a future Phase II trial with optimized dosing and combination therapies would be expected to provide progress in antitumor response.

## Figures and Tables

**Figure 1 cancers-15-03627-f001:**
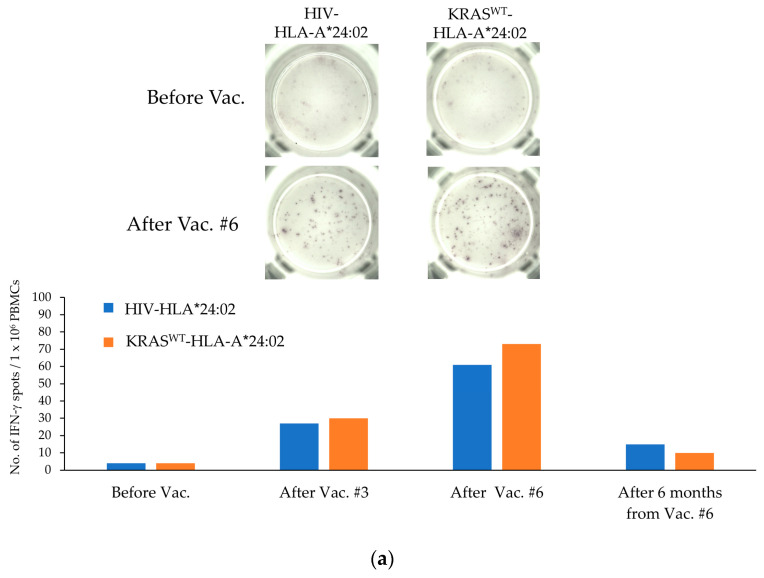
Detection of immune responses to cancer antigen candidate peptides. (**a**) Detection of IFN-γ-producing cells by ELISpot assays using PBMCs and KRAS^WT^-HLA-A*24:02 of six doses (after one course) of the HPL-APC vaccine in patient 2. HIV peptides compatible with each patient’s HLA typing was used as negative controls. Immune response was observed before and at three and six doses after HPL-APC vaccination and 6 months after completion of one course. (**b**) ELISpot assays using PBMCs and KRAS^WT^-HLA-A*33:03 after one course of HPL-APC vaccination in patient 2. (**c**) Helper peptides targeting KRAS^WT or G12D^ compatible with HLA class II in patient 2. (**d**) KRAS^G12D^-HLA-A*11:01 in patient 3. (**e**) SMAD4^WT^-HLA-A*31:01 in patient 3. (**f**) HLA class II-compliant SMAD4^G365D^, KRAS^WT or G12D^ targeted helper peptides in patient 3. (**g**) KRAS^G12D^-HLA-A*02:06 in patient 5. (**h**) Helper peptide targeting HLA class II-compatible TP53^E258K^, KRAS^WT or G12D^ in patient 5.

**Figure 2 cancers-15-03627-f002:**
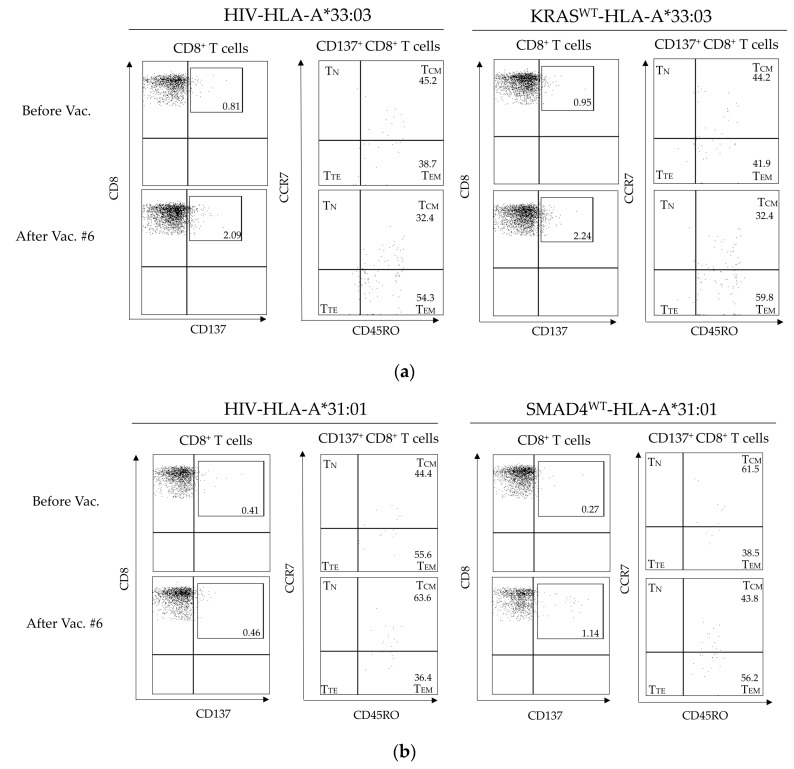
CD137 expression on CD8^+^ T cells stimulated with KRAS^WT^-HLA-A*33:03 or SMAD4^WT^-HLA-A*31:01-compatible peptides. (**a**) CD137-positive fraction in CD8^+^ T cells stimulated with KRAS^WT^-HLA-A*33:03 peptide in patient 2 were gated, and the memory subset was further analyzed. HLA-A*33:03-compatible HIV peptide was used as a negative control. (**b**) Detection of a memory subset of CD137^+^CD8^+^ T cells by SMAD4^WT^-HLA-A*31:01 stimulation in patient 3. CD45RO^+^CCR7^+^, central memory T cells (T_CM_); CD45RO^−^CCR7^+^, naïve T cells (T_N_); CD45RO^+^CCR7^−^, effector memory T cells (T_EM_); and CD45RO^−^CCR7^−^, terminal effector T cells (T_TE_).

**Table 1 cancers-15-03627-t001:** Cancer genomic profile of patients treated with the HPL-APC vaccine.

Patient No.	Age (Years)	Sex	Disease	Cancer Genomic Profiles ^†^
2	57	M	pancreatic cancer	KRAS G12D
3	48	F	cervical cancer	KRAS G12D, SMAD4 G365D
5	75	F	pancreatic cancer	KRAS G12D, TP53 E258K

†, evaluated as a pathogenic mutation by FoundationOne CDx.

**Table 2 cancers-15-03627-t002:** Prediction of cancer antigen candidate peptides for HPL-APC vaccines.

Patient No.	HLA Type	Sequence	Peptide Name	Affinity to HLA (IC_50_ nM)
2	HLA-A*24:02	QYMRTGEGF	KRAS^WT^-HLA-A*24:02	37
	HLA-A*33:03	SFEDIHHYR	KRAS^WT^-HLA-A*33:03	11
	HLA-DRB1*07:01	KQAQDLARSYGIPFI	KRAS^WT^-HLA-DRB1*07:01	40
3	HLA-A*11:01	VVGADGVGK	KRAS^G12D^-HLA-A*11:01	172
	HLA-A*31:01	CVNPYHYER	SMAD4^WT^-HLA-A*31:01	6
	HLA-DRB1*04:01	GDRFCLDQLSNVHRT	SMAD4^G356D^-HLA-DRB1*04:01	355
5	HLA-A*02:06	LVVVGADGV	KRAS^G12D^-HLA-A*02:06	164
	HLA-DRB1*01:01	TIITLKDSSGNLLGR	TP53^E258K^-HLA-DRB1*01:01	71
2, 3, 5	HLA-DRB1*09:01	TEYKLVVVGADGVGK	KRAS^G12D^-HLA-DRB1*09:01	181

**Table 3 cancers-15-03627-t003:** Quality of the HPL-APC vaccine.

Patient No.	Yield (%)	Viability (%)	Purity (%)	Phenotype (%)				
				CD40	CD80	CD86	HLA-ABC	HLA-DR
2	23	94	93	89.9	50.5	83.7	99.9	61.7
3	26	94	96	81.4	32.1	80.2	100.0	86.7
5	38	95	91	76.2	34.1	92.6	99.8	92.0

## Data Availability

The data presented in this study are available in the article and Appendix A.

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
