# Peer review of "Clinical Trial on the Safety and Tolerability of Personalized Cancer Vaccines Using Human Platelet Lysate-Induced Antigen-Presenting Cells"

_cancers, 2023, doi:10.3390/cancers15143627_

Round 1

Reviewer 1 Report

Koya et al. carried out safety, tolerability and immunoinducibility of personalized cancer vaccines using human platelet lysate-induced antigen-presenting cells using a few patients (2 pancreatic and 1 cervical cancer patients). Albeit very limited patients used and observed immunoinducibilty of their vaccine candidate with not a desirable clinical outcome, this is an interesting proof-principle study.

Specific comments:

Authors need to give rationale for using HPL-APCs in the introduction.

Author Response

We thank you greatly for your reply and review our manuscript (Manuscript ID: cancers-2446234). We appreciate the comments provided by the reviewer that have allowed us for further improvement of our manuscript. We have carefully revised the manuscript following the reviewer's suggestion. All changes have been made in a red character.

Reviewer 1

Koya et al. carried out safety, tolerability and immunoinducibility of personalized cancer vaccines using human platelet lysate-induced antigen-presenting cells using a few patients (2 pancreatic and 1 cervical cancer patients). Albeit very limited patients used and observed immunoinducibilty of their vaccine candidate with not a desirable clinical outcome, this is an interesting proof-principle study.

Specific comments:

Authors need to give rationale for using HPL-APCs in the introduction.

  1. As reviewer’s kind suggestion, we clearly revised this sentence in lines 82–85 and 86–87 of the revised manuscript.

Reviewer 2 Report

Overall, Koya et al. provides valuable evidence to verify the immunoinducibility and tolerability of HPL-APC vaccines in three enrolled patients. However, I have the following questions and comments regarding the experiment design and the results:

1. The persistence of APC vaccines is also an important for vaccine efficacy, did the authors check the persistence of HPL-APCs after vaccination? 

2. In method for generating HPL-APCs, did the authors perform CD14+ cells enrichment firstly before differentiating them into mo-DCs? If yes, this step should be mentioned in method.

3. I noticed for Flow Cytometry analysis, the authors stained with antibodies for 60mins 4 degree which is a relatively long time for such experiments. Long time incubating with antibodies is easily to increase the non-specific signals.

4. Did the authors include any data for safety evaluation of the vaccines except the descriptions in case report?

5. In addition to IFN-r producing APCs, did the authors examine the percentages of IFN-r producing T cells or antigen-specific T cells or T cell expansion after vaccination?

6. In the case report, the authors mentioned that the HPL-APC vaccination were combined with chemotherapy, did the chemotherapy affect the efficacy of the vaccines? 

7. I noticed that some of the flow cytometry graphs shown in figure 2 have less than 30 dots, it is not convincing enough for drawing conclusions. Is it possible for authors to include more events here? 

Author Response

We thank you greatly for your reply and review our manuscript (Manuscript ID: cancers-2446234). We appreciate the comments provided by the reviewer that have allowed us for further improvement of our manuscript. We have carefully revised the manuscript following the reviewer's suggestion. All changes have been made in a red character.

Reviewer 2

Overall, Koya et al. provides valuable evidence to verify the immunoinducibility and tolerability of HPL-APC vaccines in three enrolled patients. However, I have the following questions and comments regarding the experiment design and the results:

  1. The persistence of APC vaccines is also an important for vaccine efficacy, did the authors check the persistence of HPL-APCs after vaccination? 

As reviewer’s kind suggestion, we clearly explained sentences in lines 366–368 of the revised manuscript.

  1. In method for generating HPL-APCs, did the authors perform CD14+ cells enrichment firstly before differentiating them into mo-DCs? If yes, this step should be mentioned in method.

As reviewer’s kind suggestion, we clearly revised sentence in section of 2.1. Manufacture of HPL-APCs without CD14+ monocytes selection in lines 106–111.

  1. I noticed for Flow Cytometry analysis, the authors stained with antibodies for 60mins 4 degree which is a relatively long time for such experiments. Long time incubating with antibodies is easily to increase the non-specific signals.

As reviewer’s kind suggestion, we clearly explained sentence in line 120.

  1. Did the authors include any data for safety evaluation of the vaccines except the descriptions in case report?

The evaluation items are specifically described in 2.4 (lines 136–143). In this study, the safety and tolerability of HPL-APCs vaccine were confirmed in three cases described in lines 177–180, 189–191, 192–193 and 201–202.

  1. In addition to IFN-r producing APCs, did the authors examine the percentages of IFN-r producing T cells or antigen-specific T cells or T cell expansion after vaccination?

As reviewer’s kind suggestion, we clearly revised sentence in lines 315–318.

  1. In the case report, the authors mentioned that the HPL-APC vaccination were combined with chemotherapy, did the chemotherapy affect the efficacy of the vaccines? 

As reviewer’s kind suggestion, we clearly revised this sentence in lines 369–371 of the revised manuscript.

  1. I noticed that some of the flow cytometry graphs shown in figure 2 have less than 30 dots, it is not convincing enough for drawing conclusions. Is it possible for authors to include more events here? 

As reviewer’s kind suggestion, we clearly revised this sentence in line 347 of the revised manuscript. We mentioned the convincing of the results from figure 2 in lines 347–350.